# Immune-Mediated Retinal Vasculitis in Posterior Uveitis and Experimental Models: The Leukotriene (LT)B4-VEGF Axis

**DOI:** 10.3390/cells10020396

**Published:** 2021-02-15

**Authors:** Malihe Eskandarpour, Miles A. Nunn, Wynne Weston-Davies, Virginia L. Calder

**Affiliations:** 1UCL Institute of Ophthalmology, University College London, London EC1V 9EL, UK; v.calder@ucl.ac.uk; 2Akari Therapeutics Plc, London EC1V 9EL, UK; miles.nunn@akaritx.com (M.A.N.); wynne.weston-davies@akaritx.com (W.W.-D.)

**Keywords:** retinal vasculitis, uveitis, EAU, VEGF, LTB_4_, BLT1, inflammation, neovascularisation

## Abstract

Retinal vascular diseases have distinct, complex and multifactorial pathogeneses yet share several key pathophysiological aspects including inflammation, vascular permeability and neovascularisation. In non-infectious posterior uveitis (NIU), retinal vasculitis involves vessel leakage leading to retinal enlargement, exudation, and macular oedema. Neovascularisation is not a common feature in NIU, however, detection of the major angiogenic factor—vascular endothelial growth factor A (VEGF-A)—in intraocular fluids in animal models of uveitis may be an indication for a role for this cytokine in a highly inflammatory condition. Suppression of VEGF-A by directly targeting the leukotriene B4 (LTB4) receptor (BLT1) pathway indicates a connection between leukotrienes (LTs), which have prominent roles in initiating and propagating inflammatory responses, and VEGF-A in retinal inflammatory diseases. Further research is needed to understand how LTs interact with intraocular cytokines in retinal inflammatory diseases to guide the development of novel therapeutic approaches targeting both inflammatory mediator pathways.

## 1. Introduction

Non-infectious posterior uveitis (NIU) is defined as inflammation of the uveal tract. Approximately 10–15% of blindness affecting young and middle-age people in the West is caused by uveitis [1]. Therapies for posterior uveitis are limited to corticosteroids, cyclosporine and tumour necrosis factor (TNF)-blocking agents as well as off-label use of methotrexate, azathioprine and mycophenolates [2,3]. However, serious side effects (e.g., glaucoma, cataracts, liver failure and immunosuppression) can arise from long-term use of these treatments [4]. Therefore, there is an urgent need for new therapeutic interventions which are more specific, and which cause fewer side effects. To achieve this goal, an improved understanding of disease pathogenesis is needed in particular for the identification of key mediators that may be therapeutic targets.

Experimental Autoimmune Uveitis (EAU) is an animal model of NIU which is induced by CD4^+^T cell immune responses to retinal antigens. The main features of EAU are retinal and/or choroidal inflammation, retinal vasculitis, photoreceptor destruction and loss of vision, all of which resemble pathological features in human NIU. The model is well-established and commonly used for investigating fundamental mechanisms of retinal immunopathogenesis. EAU can be induced in many species but is most commonly induced in mice and rats by active immunisation with retinal antigens: interphotoreceptor binding protein (IRBP) for mice, and retinal soluble antigen (SAg) for rats [5,6,7]. 

Whilst angiogenesis and ischemia are not a common feature of uveitis, vascular inflammation, tissue damage and retinal complications are involved in vision loss [8,9]. Neovascularisation is a major problem in ocular diseases affecting the retina. The underlying stress leading to the induction of new vessel growth is mainly a result of ischemia but also inflammation, which causes the induction of angiogenic factors, including vascular endothelial growth factor A (VEGF-A), by local tissue resident cells [10]. VEGF-A has been detected in intraocular fluids during EAU and in some cases of NIU [11,12,13]. In this review, VEGF-A is referred to as VEGF unless indicated otherwise.

VEGF expression in inflammatory or ischemic conditions has been of great interest due to the recent success of therapeutic targeting of angiogenic pathways with anti-VEGF antibodies in ocular vascular diseases [14,15]. However, based on experimental approaches, a high level of VEGF does not always lead to angiogenesis and new vessel formation [16]. The reason might be that additional microenvironmental conditions/mediators are needed for angiogenesis. Such mediators may include cytokines, growth factors and bioactive lipids which all have the potential to alter tissue homeostasis, especially in the immune-privileged posterior segment in which the ocular microenvironment provides protection against retinal immune insults. Hence, there is a need to understand the retinal vascular pathways involved in uveitis and in EAU progression.

Leukotrienes (LT) have been shown to be involved in VEGF expression and in inducing angiogenesis whilst causing vascular inflammation [17,18,19]. Recent findings have demonstrated clinical efficacy of targeting the high-affinity LTB_4_ receptor (BLT1) pathway in EAU by applying a BLT1 antagonist [20] and by capturing LTB_4_ which prevented tissue damage and retinal complications [21] and suggest that the LTB_4_ pathway is a promising new therapeutic target in intraocular inflammatory diseases. Based on additional preliminary data, we hypothesise that VEGF expression and function in retinal inflammation during EAU is LTB_4_-dependent and, in this review, present further evidence to support our hypothesis.

## 2. Retinal Vasculitis in Non-Infectious Intraocular Inflammatory Disease: Uveitis

Vascular inflammation or vasculitis, where the blood vessels are inflamed, slows the blood flow to the organs and tissues. Vascular inflammation refers to a set of clinical criteria that can be used for assessing an inflammatory disease, while vasculitis describes a disorder with a set of symptoms related to tissue/organ vessel inflammation. Vasculitis can affect a wide spectrum ranging from small vessels to large arteries or veins [22]. Within the eye, retinal vasculitis is a pathological feature of sight-threatening disease which can be detected clinically and confirmed with the help of retinal fundoscopy and retinal fluorescein fundus angiography. Active retinal vascular disease is characterised by exudates around retinal vessels resulting in sheathing or cuffing of the affected vessels, which may be segmental or confluent [23,24,25]. Vascular disease is usually associated with a systemic disease and develops mainly secondarily to endothelial cell damage and dysfunction. Studies have reported that VEGF-A and VEGF-B increase collateral vessel development in ischemic animal models and in inflammatory lesions due to endothelial damage [26,27].

NIU is considered to be an immune-mediated or autoimmune disease where, in most cases, the underlying autoimmunity and potential role of infection in other parts of the body are unclear. Behçet’s disease (BD) is a chronic and recurring inflammatory disease with a multisystemic vasculitis, which is associated with vascular endothelial dysfunction. Ocular involvement in BD occurs in approximately 70% of patients [27] with the same clinical manifestations as NIU [24]. Çekmen and colleagues found that plasma VEGF-A levels were significantly higher during active BD and in ocular BD than in healthy controls [27]. Retinal vasculitis in NIU involves vessel leakage leading to retinal swelling, exudation, and macular oedema. Macular oedema is a significant contributing factor for poor vision [28]. Vasculitis affecting the peripheral retinal vasculature can cause retinal oedema, intraretinal haemorrhages and haemorrhagic infarction of the retina [29]. Ischemic retinal vasculitis may also be secondary to BD and multiple sclerosis [30].

Vitreous and intra-retinal cellular infiltrates are characteristic of infectious processes but, in the absence of these, they are pathognomonic of NIU. These transient white patches indicating retinitis, often with small adjacent haemorrhage, are almost always seen in patients with active disease. Typically, they are silent on fundus fluorescein angiography [25]. These features are also the clinical criteria used to score EAU in mice. Several studies in NIU have demonstrated a central role for CD4^+^T lymphocytes (mainly Th17 and Th1 cells) and their associated cytokines in its pathogenesis [31,32,33]. This has been supported with data from the mouse model of uveitis, EAU. In addition to CD4^+^T cells, neutrophils, recruited by cytokines and chemokines, also play an important role both in inflammation and in the resulting retinal tissue damage due to their production of superoxide anion (O2^−^) via NADPH oxidase (Nox2). Resident and infiltrated macrophages play vital roles both as effectors of innate immunity and inducers of acquired immunity. Based on EAU studies, they are major effectors of tissue damage in uveitis and are also considered to be potent antigen-presenting cells. In both phenotypic and functional terms, macrophages have enormous heterogeneity and are thought to have become specialised as tissue-resident macrophages in response to the retinal microenvironment. Cytokines tumour necrosis factor-alpha (TNFα), interleukin (IL)-6 and monocyte chemoattractant protein-1 (MCP-1) can be produced by activated macrophages during uveitis and are involved in the associated inflammatory responses [34].

One of the consequences of retinal inflammation is macular oedema during which the permeability of retinal blood vessels increases and subsequent leakage results in pooling of vascular contents in the macular tissue. Vitreous haemorrhage is also a consequence of retinal inflammation where fragile and immature blood vessels are exposed to the inflammatory microenvironment and become damaged, further increasing the vascular leakage [35]. Persistent stress on retinal tissues due to inflammation and oedema results in retinal detachment, and eventual vision loss [36].

## 3. Inflammatory Mediators Involved in Retinal Vasculitis

Retinal vascular diseases with multifactorial pathogeneses share several key pathophysiologic aspects including vascular permeability and neovascularisation. Vessel permeability, recruitment of inflammatory cells (in some cases) and retinal damage are observed during disease progression in which inflammatory mediators including cytokines and lipid mediators play critical roles. Endothelial cells are activated by proinflammatory mediators generated during inflammation, and the progression of endothelial inflammatory responses is linked to angiogenesis [37,38].

Proinflammatory cytokines produced by activated immune cells especially effector CD4^+^T cells (Th1, Th17) are important inflammatory mediators in driving NIU and in EAU [39]. Soluble inflammatory mediators (cytokines) such as IL-6, IL-17, IL-22, IL-23, interferon γ (IFNγ) and TNFα are all considered to be important contributors to disease development and progression [34,40]. The activation of the key signalling pathways in ocular inflammation—JAK-STAT (Janus kinase-signal transducer and activator of transcription protein) (JAK1, JAK2, STAT3) and mitogen-activated protein kinase (MAPK)—lead to the production of both inflammatory cytokines and VEGF [35].

LTs are a subset of lipid mediators that are potent enhancers of innate immune cell activity and are implicated in numerous inflammatory disorders. LTs primarily function to recruit immune cells including neutrophils and macrophages into areas of tissue damage or inflammation and help promote the production of inflammatory cytokines. LTB_4_ and cysteinyl leukotrienes (CysLTs) are two particularly important lipid mediators which have been detected in inflammatory eye diseases [17,41].

## 4. Cytokines in Retinal Vasculitis

NIU is driven by effector CD4^+^T cells and activated myeloid cells. Inflammatory cytokines such as IL-17, IL-6, IL-23, TNFα, IFNγ, IL-8, GM-CSF (granulocyte-macrophage colony-stimulating factor), and IL-1β have all been reported to be involved in retinal vascular inflammation [1,34]. The main cytokines involved in the pathogenesis of ocular vasculitis and neovascularisation are IL-17, IL-23, IL-6, IFNγ, TNFα, and VEGF whose contribution to disease pathology is reviewed below:

**IL-17** is mainly produced by Th17 cells but is also produced by *γδ*-T cells and Thy-1(+) innate lymphoid cells [42]. This cytokine is associated with various physiological and pathological processes such as inflammatory responses and angiogenesis of ocular diseases. There are several mechanisms proposed for IL-17-mediated effects in ocular disease. IL-17-mediated retinal and choroidal neovascularisation (CNV) is thought to be due to cytoskeleton remodelling, regulation of VEGF and related cytokines [43]. In addition, IL-17 signalling also drives the expression of metalloproteases that can cause tissue injury, and of chemokines that recruit neutrophils to the site of inflammation [44]. Although clinical trials indicate that pathological dysregulation of IL-17 plays a controversial role in ocular neovascularisation, the underlying mechanism is unknown. However, we know that IL-17 can promote angiogenesis, either directly or via stimulating VEGF production. IL-17 neutralisation decreased subretinal neovascularisation in mice overexpressing VEGF, and also significantly reduced laser-induced CNV and oxygen-induced retinal neovascularisation [45,46].

One study demonstrated that IL-17 neutralisation decreased ocular neovascularisation by promoting M2- and mitigating M1 macrophage polarisation. Therefore, the authors postulated that the pro-angiogenic effect of IL-17 might be through promoting M1 macrophage polarisation, which enhances neovascularisation [47]. The IL-17-induced macrophage polarisation and Müller cell injury are thought to be VEGF-dependent, and this mechanism is thought to be involved in cytoskeleton reconstruction [48].

In a corneal neovascularisation model, IL-17 promoted alkali-induced corneal neovascularisation through increased infiltration of intracorneal progenitor/inflammatory cells, as well as increased production of VEGF and IL-6 by fibroblasts and macrophages [49]. Talia et al. demonstrated that IL-17 inhibition attenuated neovascular retinopathy in the oxygen-induced retinopathy (OIR) mouse model, and inhibition of the RORγt (the Th17-associated transcription factor)/IL-17 axis resulted in decreased VEGF production by reduction of microglia and Müller cell gliosis, but with prevention of ganglion cell loss [50]. Hasegawa et al. reported that the pro-angiogenic effect of IL-17 in the CNV model was independent of VEGF, and the main sources of IL-17 were tissue-resident cells (*γδ*-T cells and Thy-1(+) innate lymphoid cells) [51].

In a non-immune context, VEGF and IL-17 receptor-C (IL-17RC) overexpression were both detected in retinal pigment epithelial cells (RPE) in the hypoxia group with neovascularisation as compared to that under normoxia conditions. Thus, IL-17RC might be considered as a significant indicator of corneal neovascularisation and RPE degeneration in in vitro conditions [52]. Inhibition of RORγt and IL-17 suppresses neovascular retinopathy [50].

Elevated levels of IL-17 have been identified in the eyes of patients with different types of intraocular inflammation including birdshot chorioretinopathy, Vogt–Koyanagi–Harada, as well as in HLA-B27 and Behçet’s uveitis. Elevated serum levels of IL-17 have also been identified in NIU, and increased Th17 cell levels were associated with active disease in ocular BD [53,54,55].

**IL-23** is an important cytokine for promoting the proliferation and maintenance of Th17 cells, although it is not necessary for the initial stages of Th17 differentiation [5]. The IL-17/IL-23 axis links adaptive and innate immunity in some inflammatory diseases including uveitis and psoriasis [56,57,58]. In a proteomic study of vitreous samples from NIU, an increase in IL-23 was detected [59]. In addition, elevated IL-23 has been observed in the serum and the supernatants of peripheral blood mononuclear cells from patients with active Vogt‒Koyonagi‒Harada and ocular BD when compared with patients with inactive uveitis and normal control subjects [60]. Increased serum levels of IL-23 have also been identified as a risk factor for developing anterior uveitis in patients with spondyloarthritis (an inflammatory rheumatic disease) [61]. Interleukin-23/IL-23R signalling activates the intracellular JAK/STAT signalling pathway (using JAK2, Tyk2, STAT3, and STAT4) in Th17 cells, resulting in the production of IL-17 and IL-22, which can drive chronic tissue inflammation. Interleukin-23 is expressed by dendritic cells, macrophages, and endothelial cells [57]. The impact of IL-23 on vascular inflammation is through IL-17. Recently, it has been shown that an IL-23-independent induction of IL-17 from *γδ*-T cells and innate lymphoid cells promotes experimental intraocular neovascularisation [51].

**IL-6** is involved in the differentiation of CD4^+^ T cells into Th17 cells. The levels of IL-6 are elevated in other ocular vascular diseases such as retinal vein occlusion and diabetic macular oedema. IL-6 is a major threat to ocular immune privilege and its level is extremely high in inflamed eyes where it is produced by parenchymal cells. The synthesis and secretion of IL-6 are coordinated with synthesis and secretion of pro-inflammatory IL-1 and TNFα, which enhance inflammation and ocular tissue damage [62,63]. The breakdown of the blood–ocular barrier in an endotoxin-induced model of acute anterior uveitis has been correlated with enhanced levels of TGFβ and IL-6. In addition, it has been shown that IL-6 functionally antagonises TGFβ and abolishes immune privilege in eyes with endotoxin-induced uveitis [64]. While Cohen et al. in 1996 showed that IL-6 induces VEGF expression and its function in angiogenesis [65], Rojas et al. later demonstrated a critical role for IL-6 in mediating angiotensin-induced retinal vascular inflammation and remodelling in which upregulation of VEGF is thought to be involved [66]. Elevated levels of IL-6 have been detected in ocular BD, Vogt–Koyanagi–Harada, sarcoidosis, idiopathic uveitis, acute retinal necrosis and in HLA-B27-mediated anterior uveitis when compared with controls. IL-6 also plays a role in uveitic complications such as neovascularisation and macular oedema (all studies cited here [67]). The efficacy of IL-6 inhibitors was supported in various clinical studies including a phase 2 clinical trial (STOP-Uveitis, tocilizumab) [66].

**IFNγ** is a cytokine with important roles in tissue homeostasis, immune and inflammatory responses. It is primarily produced by immune cells (CD4^+^T cells, NK cells) involved in innate and adaptive immunity. Macrophages are a major physiological target for IFNγ action and its receptors (IFNγR1 and IFNγR2) can be expressed by most cell types. IFNγ signalling activates the JAK-signal transducer and STAT1 pathway to induce the expression of associated genes [68]. The IFNγ-producing Th1 subset is responsible for the pathology of uveitis [69]. IFNγ mRNA is upregulated in R14 (IRBP_1169–1191_)-specific T cells in rat EAU causing relapsing uveitis [70], and increased levels of IFNγ-producing CD4^+^T cells were demonstrated within NIU retinal tissues in comparison with non-inflamed controls [71].

The majority of studies analysing the role of IFNγ in disease pathogenesis have focussed on its effects on immune or epithelial cells. Noticeably, data suggested a paradoxical role for IFNγ as a protective or a proinflammatory cytokine [72]. IFNγ attenuates the differentiation of Th17 cells and osteoclasts, whereas loss of IFNγ has a protective effect in collagen-induced arthritis. IFNγ also increases the expression of apoptotic mediators and regulates inflammatory cell death by targeting necroptosis in experimental autoimmune arthritis [73]. This contradiction has been clarified in EAU as the production of IFNγ early in the response—mostly from innate immune cells—has a protective function and inhibits the subsequent adaptive responses to the uveitogenic antigen [72], while in late EAU it functions as a pathogenic cytokine. In addition, IFNγ exerts potent activities on the vasculature, increasing the permeability of endothelial cell monolayers. Blockade of IFNγ by a specific antibody resulted in increased vessel density and reduced vessel permeability in IBD models, a chronic inflammatory disorder which is heavily vascularised. Langer at el. found that IBD-associated vascular barrier defects are predominantly due to IFNγ–induced deactivation of VE-cadherin in adherens junctions which supports its immunopathologic role [74].

**TNFα** is a cytokine involved in the pathogenesis of inflammatory, oedematous, neovascular and neurodegenerative disorders. TNFα exists in a soluble and membrane-bound form and is a homotrimeric cytokine [75]. Elevated levels of TNFα have been detected in the eyes and serum of patients affected by uveitis [76]. Binding of TNFα to its two receptors TNFR1 and TNFR2 can result in multiple downstream effects including inflammation, cell proliferation, apoptosis, tissue degeneration, host defence and necroptosis [77,78]. Briefly, upon TNFα binding to TNFR1, a large complex of proteins assembles at the receptor and signals to the cytoplasm via phosphorylation with subsequent proteasome-mediated degradation of inhibitor of nuclear factor-κB (IκB), an inhibitor of NF-κB. Once free of its inhibitor, NF-κB enters the nucleus and turns on the transcription of many different pro-inflammatory cytokines. Nagineni et al. demonstrated that inflammatory cytokines, including TNFα, increase the secretion of VEGF-A and -C by human RPE cells and choroidal fibroblasts [79].

Anti-TNFα therapy is effective as a treatment for severe sight-threatening uveitis, both in patients with ocular BD and in idiopathic endogenous uveitis. It achieves clinical management with biologics such as infliximab, adalimumab, golimumab, and certolizumab-pegol, or with receptor fusion protein, etanercept. Although anti-TNFα drugs are approved for many other chronic immune-mediated inflammatory diseases, they are still used off-label for non-infectious posterior uveitis, with the exception of adalimumab which has been approved for this indication [76,80,81]. Although patients benefit from anti-TNFα drugs, due to the complex formation and often opposing effects of TNFα binding to its receptors, they can cause adverse effects. A greater understanding of the downstream effects of TNFα has suggested that agents that only target TNFR1, without targeting TNFR2, may have fewer side effects [77].

**VEGF** is present in four isoforms (VEGF-A, -B, -C, and -D), which signal through three transmembrane tyrosine kinase receptors (VEGFR-1, -2, and -3). VEGF-A is a potent cytokine capable of modulating angiogenesis and vasculogenesis and is produced by many different cell types such as macrophages, monocytes, neutrophils, and endothelium. VEGF-A has an autocrine function as a survival factor on retinal pigment epithelial cells (RPE) cells, and it is needed to maintain the vessels of the choriocapillaris [35]. VEGF-B mediates embryonic angiogenesis. VEGF-C and VEGF-D are ligands for VEGFR-3 which mediates lymphangiogenesis. Like VEGF, IL-8/CXCL8 (C-X-C motif chemokine ligand 8) is also an angiogenic and anti-apoptotic survival factor for RPE cells, and both cytokines can induce neovascularisation in pathogenic conditions [82]. Despite the positive effect of VEGF blockade on the vascular leakage and regression of neovascularisation, severe damage as seen in the development of geographic atrophy and poor vision has been observed in age-related macular degradation (AMD) patients after extended treatment with anti-VEGF antibodies [83].

## 5. Neovascularisation in Inflammatory Conditions

Angiogenesis is not typically associated with NIU and EAU models. However, the major angiogenic factor VEGF has been detected in the retina during EAU [12] with no sign of neovascularisation. Although in a rat model of EAU, applying SAg peptide, neovascularisation has been reported in the later stages of disease [84]. Successfully treating uveitic complications such as cystoid macular oedema, CNV and retinal neovascularisation by applying intravitreal anti-VEGF therapies, bevacizumab and ranibizumab are evidence of a role for VEGF in the pathogenesis of these complications [11,14,15].

The association of chronic inflammation with angiogenesis is well-established and an interplay is described in the pathogenesis of major retinal diseases. These connections between inflammation and angiogenesis are reflected by the efficacy of steroids in diabetic retinopathy and of anti-VEGF in some uveitis patients. However, there are retinal disease processes in which there are increased levels of VEGF in the absence of retinal neovascularisation. Examples of these diseases in man include ocular melanoma, non-proliferative diabetic retinopathy, and a variety of non-ischemic retinal disorders [16]. There are several possible reasons that could account for the absence of neovascularisation despite VEGF upregulation in various disease processes. It is possible that one or more additional angiogenic growth factors is necessary for retinal neovascularisation to occur and is missing in the disease processes [12]. One example is in the Lewis rat model of EAU in which high levels of VEGF in the retina correlated with a high level of TGFβ1, TGFβ2 but no neovascularisation whilst rats under hypoxic conditions developed neovascularisation accompanied with increased levels of VEGF [12]. TGFβ is one of the possible inhibitors of neovascularisation as both TGFβ1 and TGFβ2 suppress vascular endothelial cell proliferation [85]. Another explanation could be that there may not be an upregulation of VEGF receptors in the retinopathies that are not associated with neovascularisation [12].

Modifications of retinal endothelial cell phenotypes have been described which correlated with inflammatory activity [86,87]. As it is well-established that blood–retinal barrier dysfunction occurs prior to any structural damage in EAU [88], it is likely that soluble factors such as LTs and VEGF play a role in vascular permeability and vascular dysfunction. A variety of inflammatory mediators are released as EAU develops and some of these factors, such as IL-1, IL-6, IL-17 have been shown to induce VEGF in other systems [70] and these may operate in a similar fashion in EAU.

## 6. Retinal Damage Associated with Disease Severity

Neovascularisation at the site of inflammation fuels the ongoing inflammatory process. In several chronic diseases affecting the retina (e.g., diabetic retinopathy, retinal vein occlusion, AMD and chorioretinal vein occlusion), neovascularisation is a major problem. Pathological retinal neovascularisation is characterised by leaky and tuft-like vessels, which are associated with retinal exudates and haemorrhage, leading to retinal damage, retinal detachment, or both [89]. The underlying stress leading to the induction of new vessel growth is mainly ischemia, which causes the induction of VEGF by local cells. However, the impact of inflammation cannot be ruled out since VEGF has been detected in ocular BD and uveitic macular oedema [11,14]. Besides the hypoxia-regulated growth factors, other non-oxygen-regulated growth factors acting in this function include the Tie (tyrosine kinase with immunoglobulin-like and EGF-like domain)1-Tie2 receptors, the Tie2 ligand angiopoietin 2. Several cytokines, hormones, and growth factors also regulate VEGF gene expression, leading to the release of VEGF [90]. VEGF is the most specific mitogenic factor for vascular endothelial cells. Phenotypical modifications in retinal endothelial cells during EAU progression have been proposed to increase the level of angiogenic factors including VEGF [87]. Its role in the pathogenesis of uveitic complications such as choroidal-retinal neovascularisation has been described [91] and is further supported by the proven efficacy of intravitreal anti-VEGF therapies in uveitis [14,15]. It has been demonstrated that a prolonged duration of the inflammatory condition involves cytokines and IL-6 and TNFα are strongly associated with retinal damage and disease severity [92].

## 7. EAU Angiogenesis

Chorioretinal neovascularisation is not a common feature of NIU but is a severe late sequela of the disease and it counts as one of the complications in NIU [8,9]. Our group has shown in a mouse model of uveitis a high level of VEGF in the vitreoretinal space during disease progression which was associated with retinal structural damages [21] and in line with findings from rat EAU when chorioretinal neovascularisation appeared during disease [93].

In C57BL/6J and B10RIII mice, EAU was induced by immunising with IRBP_1–20_ peptide and IRBP_161–180_ peptide, respectively, in PBS emulsified with CFA (Complete Freund’s Adjuvant) supplemented with *Mycobacterium tuberculosis*. Mice also received pertussis as previously described [39]. EAU in C57BL/6J mice appears as a chronic form of disease which peaks on day 21 post-immunisation, while EAU in B10RIII mice shows an earlier peak of disease on day 14 post-immunisation. Vitreoretinal fluids collected from EAU in the B10RIII model on days 14 and 19 were analysed and a significant level of VEGF was detected in EAU cases which was accompanied by severe retinal structural damages detected by fundoscopy (Figure 1).

Lewis rat EAU, induced by the retinal antigen SAg (peptide), presents with only one clinically observable acute course of intraocular inflammation without any further relapses but results in the formation of choroidal–retinal neovascularisation, visible by histology beginning 4 weeks post immunisation [94]. PP-001, a dihydroorotate dehydrogenase (DHODH) inhibitor, preferentially targets T cells and suppresses their proliferation as well as cytokine secretion. A recent study reported that neovascularisation in rat uveitic eyes was induced by VEGF secreted by the intraocular CD4^+^T cells, although VEGF secretion by T cells is not a well-recognised phenomenon. However, the group reported that PP-001 specifically targeted intraocular T cells and their VEGF secretion and subsequent choroidal–retinal neovascularisation induction without disturbing the homeostasis of basal VEGF-secretion of RPE cells that maintains the integrity of the choriocapillaris [94,95]. Retinal neovascularisation can develop during retinal inflammation similar to the retinal vasculitis observed in rat EAU without angiographic signs of ischemia or alterations of the RPE or choroid [9]. The SAg-autoreactive T cells also produced a high level of IL-6 and IL-10 compared to the controls. Those VEGF-secreting autoreactive T cells remained within the retina even after resolution of disease [93]. Suzuki et al. have shown that human immunoregulatory Foxp3^high^ CD4^+^(Treg) cells express VEGF receptor 2, and if rat Treg cells also express these receptors, then they could be fuelled by the VEGF secreted by retinal cells, thus preventing further relapses of EAU [96].

It has been speculated that in retinal inflammation involving vasculitis and neovascularisation without angiographic signs of ischemia or alterations of RPE or choroid, VEGF may be produced by inflammatory T cells that persist in the eye for an extended time and thus lead to growth of new vessels. Therefore, it has been suggested that targeting T cells whilst sparing the RPE could be considered as a therapeutic option for treating uveitis patients [94].

## 8. Potential Treatments for Retinal Vasculitis by Targeting Angiogenic Pathways

By gaining an in-depth understanding of the interactions between soluble factors like growth factors, lipid mediators and angiogenic factors, it may be possible to develop new strategies for the treatment of these conditions. Literature searches for further understanding the connections between lipid mediators and angiogenesis provided evidence for the effect of inhibiting LT pathways.

Apart from targeting VEGF, targeting VEGF-associated regulators has also shown great promise in controlling and suppressing ocular retinal vascular inflammation and relevant complications. Activation of the Tie2 receptor pathway is a new therapeutic approach, which leads to downstream signalling that promotes vascular health and stability and decreases vascular permeability and inflammation. A newly-developed drug, AXT107, a non-RGD (Arg-Gly-Asp) 20-mer α_v_β_3_ and α_5_β_1_ integrin-binding peptide, has been shown to inhibit VEGF signalling and stimulate Tie2 through the conversion of Angiopoietin 2(Ang2) into an activator [91]. These translational approaches help us to understand the underlying inflammatory pathways of ocular vascular diseases [97]. Ang2 is a mediator of inflammation which potentiates the activity of TNFα. It is rapidly secreted from inflamed endothelial cells and antagonises the vessel-stabilising activities of the Tie2 receptor by blocking its interaction with Angiopoietin 1(Ang1). The reduction in Tie2 signalling results in the weakening of retinal endothelial cell tight junctions and the subsequent leakage of plasma proteins that promote inflammatory processes, resulting in extravasating leukocytes. Ang2 also sensitises endothelial cells to TNFα, enhancing their expression of adhesion molecules and stimulating angiogenesis and vascular remodelling [91]. Moreover, Tie2 activation appears to directly regulate NF-κB-mediated inflammation through interactions with the A20 binding inhibitor of NF-κB 2 (ABIN2) [98]. In addition, Faricimab is a novel bispecific antibody designed for intravitreal use in the treatment of diabetic eye disease, which simultaneously binds and neutralises Ang-2 and VEGF-A [97].

## 9. Lipid Mediators in Retinal Vasculitis

Lipid mediators are crucial molecules in regulating inflammation that can act at the different stages of the inflammatory process: initiation, maintenance, resolution, and the return to homeostasis. They regulate inflammation in a spatial and temporal fashion [99].

Lipid mediators include pro-inflammatory eicosanoids and specialised pro-resolving mediators (SPM) that coexist in tightly regulated homeostasis necessary for permitting immune responses and then returning to baseline. Because of their potent bioactivities and broad impact on multiple cellular functions, their balance in cell and tissue homeostasis is critical [99,100]. Lipid mediators are produced from polyunsaturated fatty acids and play broad functions in the body. They include prostaglandins (PGs), LTs and lysophospholipids (e.g., sphingosine 1-phosphate) and are derived from arachidonic acid (AA) through two steps catalysed by 5-lipoxygenase (5-LOX). Lipid mediators function in normal host defence and play roles in inflammatory diseases, especially in allergic responses [99,101,102] (Figure 2). LTs are bioactive mediators which do their biological functions through binding to G-protein-coupled receptors (GPCRs). Different LT receptor subtypes exhibit unique functions and expression patterns. LT receptors BLT1 (high affinity) and BLT2 (low affinity) are activated by LTB_4_, whereas CysLT1 and CysLT2 receptors are activated by CysLTs including LTC_4_, LTCD_4_ and LTE_4_ [103].

The critical step in the biosynthesis of lipid mediators is the release of the fatty acid arachidonic acid (AA) mainly from membrane phospholipids by phospholipase A2 enzymes. Thus, modulation of phospholipase A2 enzymes and concomitant fatty acid release as the lipid mediator may be subject to regulation by microenvironmental sensors [17]. The ability to generate LTs is mainly restricted to innate immune cells such as neutrophils, monocytes, macrophages, eosinophils and mast cells that express the required biosynthetic enzymes. Those cells are capable of generating high levels of pro-inflammatory lipid mediators but also the inflammation-resolving lipid mediators or SPMs. SPMs are signalling molecules formed in cells by the metabolism of polyunsaturated fatty acids by one or a combination of lipoxygenase, cyclooxygenase, and cytochrome P450 monooxygenase enzymes [100]. Pre-clinical studies, primarily using animal models and human tissues, demonstrate a role for SPMs in orchestrating the resolution of inflammation. Resolvins and protectins are examples of those resolving lipid mediators [101,102]. LTs are involved in immune regulation, self-defence, and the maintenance of homeostasis in living systems with pivotal roles in acute and chronic inflammatory diseases. They have a significant impact on inflammatory and immune-related diseases including asthma and allergic rhinitis, rheumatoid arthritis, autoimmune diseases, atherosclerosis and cancer [104,105,106]. PGs mediate inflammation-related processes such as vasodilatation as well as increased microvascular permeability [102].

## 10. Leukotriene–Cytokine Associations in Retinal Vasculitis

LT signalling is involved in the pathogenesis/pathophysiology of various diseases and 5-LOX products, including LTB_4_ have been shown to promote angiogenesis and to have a positive regulatory effect on vasculogenesis [19]. It has been reported that M2 macrophages recruited to the injured retina make a critical contribution to the pathogenesis of CNV in wet AMD. Sasaki et al. demonstrated BLT1-expressing macrophages are recruited to the periphery in laser-induced CNV and are involved in the pathogenesis of AMD, and a BLT1 deficiency ameliorates progression of CNV in a mouse model of AMD. They also showed that the BLT1-expressing macrophages produced VEGF-A in a BLT1-dependent manner and that the blockade of LTB4-BLT1 signalling strongly inhibited CNV [19,84]. Damaged RPE cells release LTB_4_, which recruits immune cells such as neutrophils and inflammatory monocytes/macrophages to the injured retina. An autocrine/paracrine loop of LTB_4_ production by these cells leads to the formation of an inflammatory environment. Among these cells, M2 macrophages secrete VEGF-A, which accelerates CNV [84]. This data strongly suggests a potential novel therapeutic target via the LTB_4_-BLT1 pathway for those intraocular diseases.

We have also found a significant number of infiltrating/activated macrophages as well as activated T cells expressing BLT1 in EAU and in retinal tissues from NIU. Furthermore, detection of LTB_4_ in the vitreoretinal space during disease progression suggests an involvement of this pathway in EAU [21]. Herein, we report novel data demonstrating a significantly increased level of VEGF detected within the vitreoretinal space in EAU which correlated with the presence of haemorrhage and vascular inflammation (Figure 1).

Duah et al. demonstrated the involvement of CysLT pathways in the initiation of vessel inflammation in pathological conditions. CysLTs potentiate TNFα-induced responses in endothelial cells and increase the expression of vascular cell adhesion molecule-1 (VCAM-1) and leukocyte recruitment [17]. It has also been shown that CysLTs induce endothelial contraction and endothelial barrier disruption which increases endothelial cell permeability through a Rho kinase-dependent mechanism in ischemia-induced vascular leakage [107,108].

LTC_4_ and LTD_4_ induce calcium influx in HUVECs via CysLT2R, which leads to adherens junction disassembly and cytoskeletal rearrangements to facilitate endothelial cell retraction and increased permeability [17]. Espinosa et al. (2003) has also reported the expression of CysLT2R by retinal vasculature pericytes and endothelium in a murine model of OIR and suggested that local and selective expression of the receptor can control retinal oedema and pathological neovascularisation [109]. CysLT stimulation leads to rapid vascular leakage. Adult CysLT2R KO mice (KO receptor) subjected to local proinflammatory eye conditions (by intravitreal injections of LTs) demonstrated a decreased retinal vascular permeability compared to wild-type mice. Local pharmacological antagonism of the CysLT2R in wild-type animals could counteract inflammatory-induced vascular leakage [110]. The involvement of CysLTs in NIU and other retinal diseases’ development/progression needs further investigation.

## 11. Role of LTB_4_ Pathway in EAU and Uveitis

Our knowledge of the contribution of the LTB_4_ pathway within the retina during uveitis development/progression and in the in vivo experimental model of disease (EAU) is very limited. Technical challenges in the detection of LTs and accessibility to the posterior segment of the eye are added technical constraints. Therefore, scientists have used BLT1 KO models and receptor antagonists as a tool to study the involvement of the LTB_4_ pathway in disease development and progression. Reviewing publications on the contribution of this pathway, we have reached the conclusion that there is an involvement of LTB_4_ and BLT1 signalling in EAU and human diseased retinal tissues and have recently demonstrated the LTB_4_-dependence of retinal Th17 cells in disease progression [21].

Applying specific inhibitors or drugs to study the biology of a disease is a valuable tool. There is evidence that an LTB_4_ receptor-targeted inhibitor blocked EAU progression [20]. In addition, transferring retinal-specific autoreactive T cells from C57BL/6J mice to 5-LOX^−/−^ mice deficient in LTB_4_ expression failed to induce uveitis in recipient mice [20]. Our group has also demonstrated that the LTB_4_ pathway is important during early and later stages of EAU. We detected a significant level of LTB_4_ within the vitreous at different stages of EAU development, although not in all eyes. Detection of the LTB_4_ receptor (BLT1 with high affinity) on infiltrating/active macrophages and effector Th17 cells provides strong evidence for the contribution of this pathway in disease progression (Figure 2). In vitro studies on LTB_4_ receptor expression by antigen-specific CD4^+^T cells also confirmed the concept that, in inflammatory conditions, effector cells are able to express LTB_4_ receptors in response to activation-induced signalling pathways. Demonstrating the presence of the LTB_4_ receptor by infiltrating immune cells in human uveitis retinal samples added further evidence for a contribution of the LTB_4_-BLT1 pathway in uveitis [21].

## 12. Potential Treatment Approaches in Targeting LT Pathways in Retinal Vasculitis

Finding and targeting those pathways having the highest impact on disease outcomes is always a huge challenge. The LTB_4_ pathway is clearly active in numerous disease states including inflammatory diseases and remains a target of interest for therapeutic drug development, especially where new therapeutic indications are supported by more definitive mechanistic biology. Drugs targeting the LTB_4_ pathway are classified into 2 broad classes: antagonists of the known LTB_4_ receptors (BLT1 and BLT2) and inhibitors of the enzymes responsible for generation of LTB_4_ (e.g., TA4H and 5-LOX) [111].

Some of the BLT receptor antagonists which have reached phase 2 clinical trials are Etalocib (LY293111; Lilly), Amelubant (BIIL 284; Boehringer Ingelheim), Moxilubant (CGS-25019C, LTB-019; Novartis) and CP-105696 (Pfizer). The target diseases have been chronic obstructive pulmonary disease (COPD), asthma, rheumatoid arthritis, cystic fibrosis and cancer. Antagonist LY293111 in asthmatic patients showed that it led to a significant reduction in the number of neutrophils in BAL fluid, but failed to improve respiratory function or airway reactivity after allergen challenge [112,113,114]. However, a study examining the role of the BLT1 antagonist, CP-105696, in monkeys showed that the compound inhibited both LTB_4_-mediated neutrophil chemotaxis and upregulation of CD11b^+^ cells [99]. At least six LTA_4_H inhibitors of the current generation have entered clinical trials and reached phase 2 studies. Of the LTA4H inhibitors, Ubenimex (Bestatin; Nippon-Kayaku, Eiger Biopharmaceuticals), Acebilustat (CTX-4430 Celtaxsys) and Tosedostat (CHR-2797; CTI Biopharma) have been applied in asthma, myelodysplastic syndrome, solid tumours, active cystic fibrosis, acne vulgari, pulmonary arterial hypertension (PAH), atopic dermatitis and allergic conjunctivitis [99,112].

The well-known 5-LOX inhibitor, Zileuton, is already in clinical use for the maintenance treatment of asthma and has been tried in other inflammatory/allergic diseases including atopic dermatitis and allergic conjunctivitis. Zileuton inhibits VEGF-Induced angiogenesis, expression of cell adhesion molecules (VCAM-1, ICAM-1), TNFα secretion and production of NO [18], the latter being required for efficient angiogenesis which is synthesised by endothelial NOS. NO is produced from bone marrow-derived macrophages stimulated with IFNγ and TNFα and is thought to be produced similarly in vivo [115]. The anti-angiogenic effect of Zileuton might also relate to the activation of the large-conductance Ca2^+^-activated K^+^ (BK) channel, resulting in activation of pro-apoptotic signalling cascades. Another possible explanation is based on studies which demonstrated that Zileuton was effective in inhibiting biosynthesis of multiple AA metabolites, including 12- and 15-hydroxyeicosatetranolic acid (12-, 15-HETE) in a hamster cheek pouch model of carcinogenesis. 12-HETE is involved in mediating VEGF-induced angiogenesis [18]. Therefore, there is a possibility that the anti-angiogenic effect of Zileuton comes from preventing the 12-LOX, and not the 5-LOX, pathway thereby blocking the production of 12-HETE, the metabolite of the 12-LOX pathway [112].

A new therapeutic approach could be by using more selective inhibitors of LTB_4_ and BLT1 interactions in affected tissues/organs. Nomacopan is one example of a compound that demonstrates functional selectivity for LTB_4_ and shows a promising reduction in disease severity in a preclinical uveitis model, a mouse model of Bullous Pemphigoid [116] and is now in phase 2 clinical study for bullous pemphigoid.

## 13. Downregulation of VEGF by an LTB4 Inhibitor (Nomacopan)

Nomacopan is a recombinant protein originally derived from saliva of the soft tick *Ornithodoros moubata*. It has dual functions of specifically sequestering LTB_4_ and inhibiting complement component 5 (C5) activation [117]. Its mode of action prevents LTB_4_ interacting with its two known G protein coupled cell surface receptors (BLT1 and BLT2). It was demonstrated that nomacopan and its variants provide therapeutic benefit in experimental autoimmune uveitis [21] and demonstrated in Figure 3.

Furthermore, treating already active EAU in mice intravitreally by nomacopan, using a long-acting variant which specifically inhibits LTB_4_-BLT1 binding (L-nomacopan), suppressed disease progression and prevented structural damage to the retina with a significant reduction of effector Th17 cells and inflammatory macrophages [21]. We also demonstrated a significant reduction in the VEGF level in EAU mice treated with nomacopan intravitreally which was more robust than with an anti-VEGF mAb control (Figure 3). All of which support the importance of the LTB_4_-BLT1 pathway in this intraocular inflammatory disease and provide further evidence, suggesting the pathway as a relevant specific target for intraocular diseases. However, whether the LTB_4_-BLT1 pathway has a direct inhibitory effect on VEGF levels or an indirect effect via downregulation of retinal inflammation, needs further investigation.

## 14. Conclusions

Cytokines and inflammatory mediators are potent enhancers of immune cell functions and are implicated in numerous inflammatory disorders, including intraocular inflammatory diseases. The diversity in their cellular sources of production and biological functions adds further complexity in finding and targeting those pathways with the highest impact on disease outcomes. We have reviewed the evidence herein that LTB_4_ biosynthesis is activated in intraocular inflammatory diseases, VEGF exists in uveitic conditions and in animal models of uveitis and targeting both pathways in combination therapy may prevent further tissue damage and disease complications.

## Figures and Tables

**Figure 1 cells-10-00396-f001:**
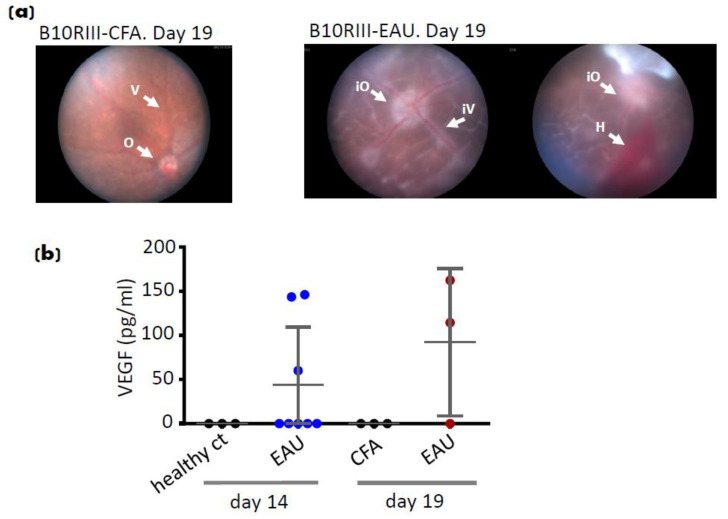
Vascular endothelial growth factor (VEGF) detection in experimental autoimmune uveitis (EAU). (**a**)**.** Fundoscopy on B10RIII EAU mice on day 14 and 19 and (**b**) VEGF levels (pg/mL) detected in vitreoretinal fluids from healthy control and EAU eyes on days 14 and CFA (Complete Freund’s Adjuvant) control and EAU eyes on day 19. Data are representative of at least 4 independent experiments with different numbers of mice (*n* = 3–6) in each group. ct = control, O = optic disk, V = vessel, iO = inflamed optic disk, iV = inflamed vessel, H = haemorrhage.

**Figure 2 cells-10-00396-f002:**
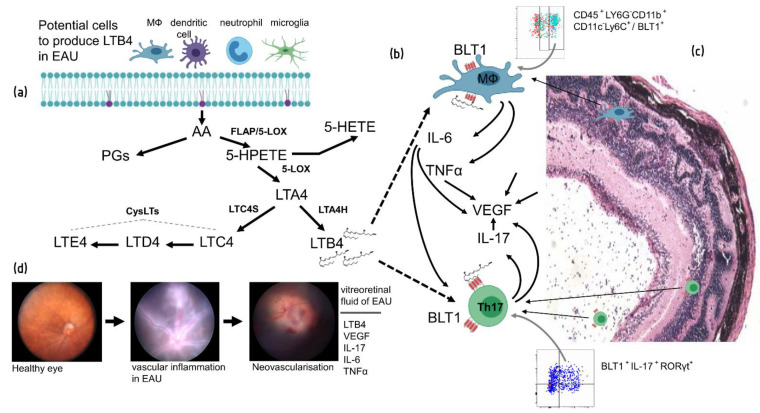
The schematic figure showing some of the microenvironmental conditions in the inflamed posterior eye in EAU with infiltrating immune cells and structural damage along with relevant inflammatory pathways. (**a**) A summary of biosynthesis of leukotriene B4 (LTB4) and cysLTs (LTC4, LTD4 and LTE4) from AA from potential immune cells in inflamed posterior chamber. (**b**) LTB4 interacts with LTB4 receptor (BLT1), a G Protein Coupled Receptor, on the infiltrating macrophages and also T cells and activates signalling pathways. Th17 cells produce cytokines including interleukin (IL)-17 and VEGF. Producing IL-6 and tumour necrosis factor-alpha (TNFα) by activated macrophages also enhances VEGF expression. The speculated signalling through LTB4-BLT1 pathway which leads to production of VEGF in Th17 and infiltrating macrophages showed by dotted lines. (**c**) The Hematoxylin and Eosin section from EAU shows histological changes in the posterior chamber of the eye with infiltrating immune cells in vitreous space and structural damages in retinal layers. (**d**) Fundoscopy of healthy eye, inflamed eye (EAU) with vascular inflammation and sign of neovascularisation during EAU progression has been shown in this figure. O = optic disk, V = vessel, iO = inflamed optic disk, iV = inflamed vessel, nV = new vessel.

**Figure 3 cells-10-00396-f003:**
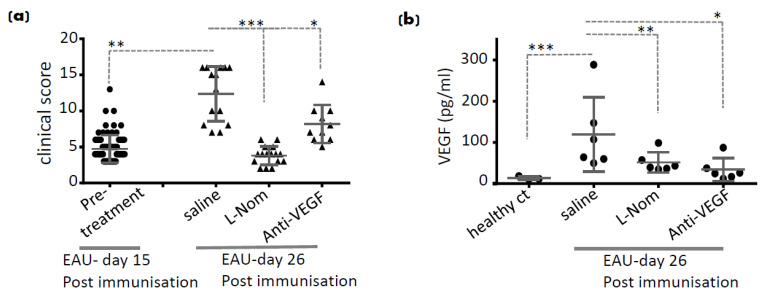
Disease suppression and VEGF downregulation. (**a**) Clinical score of EAU mice (C57Bl/6) on day 15 post-immunisation and pre-treatments. Clinical score of EAU mice at the end point on day 26 post-immunisation. Mice were treated intravitreally with 1–2 µL of L-nomacopan (L-Nom; only targeting LTB4, 20 mg/mL), saline or anti-VEGF (5 mg/kg, Ultra-LEAF™ Purified anti-mouse VEGF-A Antibody, 2G11-2A05, Biolegend) on day 15 and 18 post-immunisation. (**b**) VEGF levels (pg/mL) were detected in vitreoretinal space of corresponding EAU mice in (a) and compared to the healthy control mice. Each bar was drawn based on the mean value ± SD of each score (*n* = 8–17 mice). Unpaired *t*-test P values compared to vehicle. * *p* < 0.05, ** *p* < 0.01, *** *p* < 0.001.

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
