# Peer review of "Immune-Mediated Retinal Vasculitis in Posterior Uveitis and Experimental Models: The Leukotriene (LT)B4-VEGF Axis"

_cells, 2021, doi:10.3390/cells10020396_

Round 1

Reviewer 1 Report

The review by Eskandarpour et al. aims at demonstrating the correlation of leukotriene and VEGF pathways in uveitis. This topic is very important, and manuscript includes interesting aspects, but unfortunately it is hard to read. The chapters addressing the different topics are alternating and rather separating the topics than combining them.

The manuscript needs revision concerning typos including missing words. Some examples of incorrect references are given below. About 40% of the cited references in this review are reviews, which is not quite appropriate for a review article; the original publications should be cited instead.

Some points in detail:

  1. Authors: according to the "author contributions" (p18) Wynne Weston-Davies has not contributed to this manuscript al all, and Miles A. Nunn has only revised the paper, which also does not qualify for a coauthorship.

In addition, the authors declare no conflict of interest, but both coauthors mentioned above are employees of the pharma company Akari Therapeutics plc, and the entire chapter 5.2 is about the company's drug Nomacopan.

  1. Abbreviations: The list of abbreviations is incomplete, about half of the abbreviations used throughout the text are missing.
  2. The listing of the content is not correct; the numbering of the chapters differs from the text.
  3. Since the topics are always jumping between VEGF and LTs I would propose the following change the order of the chapters to better allow the reader to follow (present numbers of the chapters in brackets). This might also help better connecting the two main topics.
  4. Introduction
  5. Retinal vasculitis in non-infectious intraocular inflammatory disease: Uveitis (1)
  6. Inflammatory mediators involved in retinal vasculitis (2)
  7. Cytokines in retinal vasculitis (3, paragraph about VEGF shifted to the end of this chapter)
  8. Neovascularisation in inflammatory conditions (4)
  9. Retinal damage associated with disease (5.1)
  10. EAU angiogenesis (5.2)
  11. Potential treatments for retinal vasculitis by targeting angiogenic pathways? (5.3)
  12. Lipid mediators in retinal vasculitis (6)
  13. Leukotrienes-Cytokines associations in retinal vasculitis (6.1)
  14. Role of LTB4 pathway in EAU and uveitis (6.2)
  15. Potential treatment approaches in targeting LT pathways in retinal vasculitis? (6.3)
  16. Nomacopan (6.4)
  17. Conclusions
  18. References

  1. P9, line 285/284: This sentence does not really fit in here, this is one of the unlucky transitions between the topics, here trying to create a connection between VEGF and leukotrienes.
  2. P10, line 300ff : The detailed description of the lipid mediator pathways needs to be revised, this chapter would be good for a biochemistry textbook, but does not fit in this review.
  3. Fig. 1: The correlation between the pathway of the biosynthesis of LTs and uveitis is very weak and difficult to understand. What is the meaning of the three fundus pictures on the lower left? The dotted arrow from LTB4 should rather point to the BLT1-positive cells on the upper and lower right than to VEGF.
  4. P11, line 320: what is the meaning of "fatty acid substrate" and "lipid mediator substrate"? Enzymes have substrates that they are converting.
  5. P11, line 332: reference 89 does not fit here.
  6. Chapter 3.3 is mainly about LTs, while Fig. 2 is only about VEGF. This figure should be introduced in another chapter. Are the two fundus pictures at the right representing day 14 and day 19 of EAU?
  7. Chapter 4, line 379: Ref. 89 is wrong here.

Line 397: reference 97 does not really present data of a BRB dysfunction, this paper discusses the effect of "reagins" (IgE antibodies) to promote EAU induction (these days EAU was called "experimental allergic uveitis"), which has not been proven as necessary until today

  1. P14, line 409: should be "macular edema"
  2. Chapter 5, p14/15, line 433: chorioretinal neovascularization appears only after immunization with an S-Ag peptide but not with the entire SAg protein.
  3. Lines 437-440: these two consecutive sentences are repetitive.
  4. P15, lines 442/443: see above, not S-Ag, but peptide-specific
  5. P16, line 478: "PASylation should be explained at first mention.
  6. Fig. 3B and line 495: Is the reduction of the VEGF levels in the eyes of the PAS-L-Nom-treated animals due to the decreased inflammation of is there a direct effect on VEGF secretion?
  7. P17, line 505: Abbreviations: what is the Tie receptor pathway? What is RGD? A figure depicting the interactions described in lines 504-519 would be helpful, the sheer description in the text is very confusing.

In general, the topic is very interesting and ambitious, but the manuscript needs revision to make the story sound and to really show the interconnection between VEGF/neovascularization and lipid mediator pathways in uveitis.

Author Response

Reviewer 1.

We would like to thank the reviewer 1 for her/his helpful comments and suggestions. The order of the content has been changed to improve the comprehension of the paper. We have amended the manuscript where requested, adding some further data, and have changed figure legends and the text, which we have addressed as follows:

The review by Eskandarpour et al. aims at demonstrating the correlation of leukotriene and VEGF pathways in uveitis. This topic is very important, and manuscript includes interesting aspects, but unfortunately it is hard to read. The chapters addressing the different topics are alternating and rather separating the topics than combining them.

The manuscript needs revision concerning typos including missing words. Some examples of incorrect references are given below. About 40% of the cited references in this review are reviews, which is not quite appropriate for a review article; the original publications should be cited instead.

We have revised all references and added some original papers when it was needed. Most of the review papers are disease related articles which there is no other way to refer to them. In some review papers we tried to find the original papers but unfortunately in some cases the original ones were not relevant to the subject. Therefore, we kept some of the citations as they were.

Some points in detail:

  1. Authors: according to the "author contributions" (p18) Wynne Weston-Davies has not contributed to this manuscript al all, and Miles A. Nunn has only revised the paper, which also does not qualify for a coauthorship.

In addition, the authors declare no conflict of interest, but both coauthors mentioned above are employees of the pharma company Akari Therapeutics plc, and the entire chapter 5.2 is about the company's drug Nomacopan.

Response. All authors have reviewed and edited the paper, especially Miles Nunn had very useful comments and suggestions to improve the concept of the paper. We have now amended this information and also the conflict of interest accordingly.

2. Abbreviations: The list of abbreviations is incomplete, about half of the abbreviations used throughout the text are missing.

Response 2. We have now added the missing abbreviations to the list.

3. The listing of the content is not correct; the numbering of the chapters differs from the text.

Response 3. We appreciate the reviewer comment on the content list but we were unable to find any mistakes in the order. However, we have revised the list to make the contents flow more logically.

4. Since the topics are always jumping between VEGF and LTs I would propose the following change the order of the chapters to better allow the reader to follow (present numbers of the chapters in brackets). This might also help better connecting the two main topics

  1. Introduction
  2. Retinal vasculitis in non-infectious intraocular inflammatory disease: Uveitis (1)
  3. Inflammatory mediators involved in retinal vasculitis (2)
  4. Cytokines in retinal vasculitis (3, paragraph about VEGF shifted to the end of this chapter)
  5. Neovascularisation in inflammatory conditions (4)
  6. Retinal damage associated with disease (5.1)
  7. EAU angiogenesis (5.2)
  8. Potential treatments for retinal vasculitis by targeting angiogenic pathways? (5.3)
  9. Lipid mediators in retinal vasculitis (6)
  10. Leukotrienes-Cytokines associations in retinal vasculitis (6.1)
  11. Role of LTB4 pathway in EAU and uveitis (6.2)
  12. Potential treatment approaches in targeting LT pathways in retinal vasculitis? (6.3)
  13. Nomacopan (6.4)
  14. Conclusions
  15. References

 Response 4.  This is a helpful suggestion which has now been incorporated.

P9, line 285/284: This sentence does not really fit in here, this is one of the unlucky transitions between the topics, here trying to create a connection between VEGF and leukotrienes.

Response.   We agree with the comment and deleted the sentence.

5. P10, line 300ff : The detailed description of the lipid mediator pathways needs to be revised, this chapter would be good for a biochemistry textbook, but does not fit in this review.

Response 5.   The paragraph has been revised.

6. Fig. 1: The correlation between the pathway of the biosynthesis of LTs and uveitis is very weak and difficult to understand. What is the meaning of the three fundus pictures on the lower left? The dotted arrow from LTB4 should rather point to the BLT1-positive cells on the upper and lower right than to VEGF.

Response 6.   The figure 1 has been revised based on reviewers 1 and 2. Figure legend has also been changed accordingly. Since the order of figures has been changed, figure 1 is now figure 2.

7. P11, line 320: what is the meaning of "fatty acid substrate" and "lipid mediator substrate"? Enzymes have substrates that they are converting.

Response 7.   The “substrate” has been deleted in both sites. The message would be clearer now. Our attention was saying fatty acid AA and later on lipid mediators can be a substrate for enzymes including phospholipase A2.

8. P11, line 332: reference 89 does not fit here.

Response 8.   We apologise for this oversight. The correct references have now been added.

9. Chapter 3.3 is mainly about LTs, while Fig. 2 is only about VEGF. This figure should be introduced in another chapter. Are the two fundus pictures at the right representing day 14 and day 19 of EAU?

Response 9.  We have changed the order of figures and fig 2 now becomes fig 1, and in the correct place. The fundus photos refer to day 19 and we have now amended the legend for further clarification.

10. Chapter 4, line 379: Ref. 89 is wrong here.

Response 10.   This reference has been removed and the correctreferences have been added.

11. Line 397: reference 97 does not really present data of a BRB dysfunction, this paper discusses the effect of "reagins" (IgE antibodies) to promote EAU induction (these days EAU was called "experimental allergic uveitis"), which has not been proven as necessary until today

Response 11.  Thank you for spotting the error in citation. The reference has now been changed to a relevant reference, Lightman et al. 1992 who showed that BRB dysfunction occurs before retinal damage.

12. P14, line 409: should be "macular edema"

Response 12.   Thank you for spotting the typo. It has been amended.

13. Chapter 5, p14/15, line 433: chorioretinal neovascularization appears only after immunization with an S-Ag peptide but not with the entire SAg protein.

Response 13.   The “peptide” has now been added to the sentence for clarification.

14. Lines 437-440: these two consecutive sentences are repetitive.

Response 14.   The un-necessary line has been deleted.

15. P15, lines 442/443: see above, not S-Ag, but peptide-specific

Response 15.   The “peptide” has been added to the relevant places.

16. P16, line 478: "PASylation should be explained at first mention.

Response 16.   The method of PASlyation uses to prolong the half-life of the biologics, however we have deleted this part as we are not showing any data from non-PASylated forms of nomacopan, and also it is not relevant to the review topics.

17. Fig. 3B and line 495: Is the reduction of the VEGF levels in the eyes of the PAS-L-Nom-treated animals due to the decreased inflammation of is there a direct effect on VEGF secretion?

Response 17.   We have not investigated if the effect of LTB4-BLT1 pathway on VEGF expression is through a direct or indirect manner. We have added a line to clarify this in the text. 

18. P17, line 505: Abbreviations: what is the Tie receptor pathway? What is RGD?

A figure depicting the interactions described in lines 504-519 would be helpful, the sheer description in the text is very confusing.

Response 18.   Abbreviations have been added with more information to the figure 2 which now is figure 1.

Regarding adding a figure to show the Tie/Ang2 connections, we have changed the text and deleted un-necessary parts. Hope it is fluent now. Since the subject is not relevant to the main topic but it is a new intervention in inhibiting neovascularisation, it is good to have it but  it forms such a small part that we do not think another figure is required.

In general, the topic is very interesting and ambitious, but the manuscript needs revision to make the story sound and to really show the interconnection between VEGF/neovascularization and lipid mediator pathways in uveitis.

Reviewer 2 Report

Eskandarpour et al have written a review on the involvement of leukotrienes (LTs) and VEGF in retinal vasculitis and in particular non-infectious posterior uveitis (NIU). The authors have included information on various immune mediating cytokines linked to VEGF/LTs and how they play a role in retinal vascular disease changes. 

There are a few things that need to be addressed in this review:

Formatting needs to be looked at- the numbering of the sections needs to be amended both in table of contents and main text.  

Use of abbreviations needs to be consistent throughout the manuscript. It would appear some sections were written before others as some abbreviations keep being repeated eg LTs are abbreviated line 154 and then again on line 297, same for SPM. Please amend for all where appropriate. 

PG is used in main text as abbreviation for prostaglandins but in abbreviation list use PGE

Change degradation to degeneration for the abbreviation AMD 

Figure 1 needs to be labelled a, b etc to make easier for reader. Image of neovascularisation is not very clear and very dark- do the authors have a better representative image as the whole eye is not very distinguishable from background? 

Figure 2A) are the two eyes of the EAU mice for both time points (day 14 and 19) or for one time point ? It is unclear especially since the figure legend does not mention the CFA mice image. This needs to be amended. 

Fig 2B) what is the arrow with EAU indicating? There is a great variability in the samples. Is there any significance for the graph? How many eyes were studied etc as there are only 3 points for day 19 compared to 8 for day 14. The graph suggests that for some of the samples there appears to be no change in VEGF levels as they are similar to control/CFA. 

Figure 3: how many mice were used per cohort? What statistical test was used? These details should be included in figure legend. What dose of anti-VEGF was used for injection? 

The authors say that treatment with PAS-L-nomacopan prevented retinal damage- are there images that can be included to show this as it would be good to see? 

Check affiliations of authors - assume Akari Therapeutics should actually be 2 rather than 4.

Author Response

Reviewer 2:

We would like to thank the reviewer 2 for her/his helpful comments and suggestions. The order of the content has been changed to improve the comprehension of the study. We have amended the manuscript where requested, adding some further data, and have changed figure legends and the text, which we have addressed as follows:

 Comments and Suggestions for Authors

Eskandarpour et al have written a review on the involvement of leukotrienes (LTs) and VEGF in retinal vasculitis and in particular non-infectious posterior uveitis (NIU). The authors have included information on various immune mediating cytokines linked to VEGF/LTs and how they play a role in retinal vascular disease changes. 

There are a few things that need to be addressed in this review:

Formatting needs to be looked at- the numbering of the sections needs to be amended both in table of contents and main text.  

Response.   We have re-formatted the sections based on the reviewer 1 comments/suggestions and your comment here.

Use of abbreviations needs to be consistent throughout the manuscript. It would appear some sections were written before others as some abbreviations keep being repeated eg LTs are abbreviated line 154 and then again on line 297, same for SPM. Please amend for all where appropriate. 

Response.   All abbreviations have been checked and necessary changes have been done for a better consistency throughout the paper.

PG is used in main text as abbreviation for prostaglandins but in abbreviation list use PGE

Response.   We have amended this abbreviation and used PG throughout the paper.

Change degradation to degeneration for the abbreviation AMD 

Response.   Sorry that was an unforgivable mistake which has been corrected.

Figure 1 needs to be labelled a, b etc to make easier for reader. Image of neovascularisation is not very clear and very dark- do the authors have a better representative image as the whole eye is not very distinguishable from background? 

Response.  We have divided the figure 1 (now figure 2) into 4 segments and added more information to show connections between each segment. The fundoscopy images have been improved to show greater contrast. As the focus in the right photo was aimed at showing the tiny vessels, the main issue at the time of experimentation, we overlooked the importance of having a lighter background to show the periphery as well. We have no resource to repeat the experiments for an improved image.

Figure 2A) are the two eyes of the EAU mice for both time points (day 14 and 19) or for one time point ? It is unclear especially since the figure legend does not mention the CFA mice image. This needs to be amended. 

Response.  We apologise that the legend and captures were not informative. The necessary information has now been added to the fig and legend. Since we have changed the order of figures, figure 2 is now figure 1.

Fig 2B) what is the arrow with EAU indicating? There is a great variability in the samples. Is there any significance for the graph? How many eyes were studied etc as there are only 3 points for day 19 compared to 8 for day 14. The graph suggests that for some of the samples there appears to be no change in VEGF levels as they are similar to control/CFA. 

Response.   This experiment is a representative of at least 4 independent studies applying B10RIII EAU mice. In none of the studies was VEGF level detectable in all vitreoretinal fluids. Comparisons between groups didn’t give a significant difference due to the low numbers in each group. The relevant information has been added to the figure legend. Since we have changed the order of figures, fig 2 is now fig 1.

Figure 3: how many mice were used per cohort? What statistical test was used? These details should be included in figure legend. What dose of anti-VEGF was used for injection? 

Response.   This informion has been added to the figure legend.

The authors say that treatment with PAS-L-nomacopan prevented retinal damage- are there images that can be included to show this as it would be good to see? 

Response.  The data have already been published therefore we have cited the paper in this review.

Check affiliations of authors - assume Akari Therapeutics should actually be 2 rather than 4.

Response.   It has been amended.

Reviewer 3 Report

Malihe Eskandarpour et al. produced a well-written review focused on “Immune-mediated retinal vasculitis in posterior uveitis and experimental models: 1 the leukotriene (LT)B4-VEGF axis”. I consider the manuscript sufficiently complete but, in the same time, I suggest the authors to drastically improve English and correct several typos present throughout the manuscript.

Author Response

Reviewer 3:

We would like to thank the reviewer 3 for her/his helpful comments on English improvement and typos. We have amended the manuscript where requested by reviewer 1 and 2 and worked on text for a better English.

Malihe Eskandarpour et al. produced a well-written review focused on “Immune-mediated retinal vasculitis in posterior uveitis and experimental models: 1 the leukotriene (LT)B4-VEGF axis”. I consider the manuscript sufficiently complete but, in the same time, I suggest the authors to drastically improve English and correct several typos present throughout the manuscript.